# Synthesis, Redox and Spectroscopic Properties of Pterin of Molybdenum Cofactors

**DOI:** 10.3390/molecules27103324

**Published:** 2022-05-22

**Authors:** Kyle J. Colston, Partha Basu

**Affiliations:** Department of Chemistry and Chemical Biology, Indiana University-Purdue University Indianapolis, Indianapolis, IN 46202, USA; kcolsto@iu.edu

**Keywords:** molybdenum cofactor, molybdopterin, pterin, synthesis, redox, spectroscopy

## Abstract

Pterins are bicyclic heterocycles that are found widely across Nature and are involved in a variety of biological functions. Notably, pterins are found at the core of molybdenum cofactor (Moco) containing enzymes in the molybdopterin (MPT) ligand that coordinates molybdenum and facilitates cofactor activity. Pterins are diverse and can be widely functionalized to tune their properties. Herein, the general methods of synthesis, redox and spectroscopic properties of pterin are discussed to provide more insight into pterin chemistry and their importance to biological systems.

## 1. Introduction

All phyla of life harbor at least one molybdenum cofactor containing enzyme that participates in a variety of functions from global cycling of C, S, N and As to prodrug metabolism. All such enzymes contain the core molybdenum cofactor (Moco) as the central entity that catalyzes those reactions. The molybdenum cofactor is unstable outside the protein environment and it was first characterized by careful degradation studies [1]. This study revealed the complicated nature of the cofactor, which involves a pterin moiety and a 1,2-ene dithiolate appended to this pterin. The 1,2-ene dithiolate ligand binding to a metal is unique in biology, and only found in molybdenum and tungsten enzymes. Subsequent crystal structures have revealed the presence of a pyran ring which at times is found to be open. While the basic structure is retained in all Mo/W enzymes, a point of variability exists in the phosphate terminus. In some bacterial proteins, the phosphate group is replaced by a dinucleotide.

Structurally, the pterin molecule consists of a pyrimidine ring and a pyrazine ring fused together forming a complicated bicyclic N heterocycle. Pterin can exist in several redox states, indicated by the prefix tetrahydro, trihydro or dihydro, that denote the number of hydrogen atoms attached to the pyrazine ring. The dithiolate functionality is appended at the 6-position of the pterin, consistent with most biologically relevant pterin molecules. The pterin chemistry is quite mature and there are excellent review articles discussing different aspects [2,3,4,5,6,7,8,9,10,11,12,13,14,15,16,17]. This article focuses on the synthesis, redox, spectroscopic and computational studies of pterins as they pertain to the molybdenum cofactor and the variety of pterins that are found at the catalytic core of molybdopterin enzymes (Figure 1) [18].

## 2. Synthesis of Pterin

The total synthesis of pterin and similar cofactor precursors can pose challenging problems due to the highly specific stereochemistry and functionalized nature of many relevant bioorganic molecules. In essence, pterin is an aromatic and bicyclic compound composed of pyrimidine and pyrazine moieties (Figure 2), and either component can be used as the initial building block for pterin synthesis. In general, starting with a pyrimidine affords more flexibility in functionalization while utilizing a pyrazine ring as the initial precursor provides an alternative approach. Desired functionalization can be performed during pteridine formation or achieved after pteridine cyclization, both of which can be used for the incorporation of the pyran ring that forms the pyranopterin unit of molybdenum cofactor.

### 2.1. Gabriel-Isay Condensation

Condensation of 5,6-pyrmidinediamines with 1,2-dicarbonyl type compounds is one of the most popular methods for pteridine synthesis (Figure 1), more commonly known as the Gabriel–Isay condensation [19]. Pteridine functionalization is largely driven by the diversity of 1,2-dicarbonyl species which include but are not limited to diketones, dialdehydes and ketoaldehydes. The Gabriel–Isay condensation initializes with a nucleophilic attack of the most reactive amino group on the more electropositive carbonyl. A similar nucleophilic attack occurs between the remaining amino and carbonyl groups resulting in pyrazine formation and the dehydration of two equivalents of water. The asymmetric nature of pyrimidine can produce a mixture of isomers which is dependent on the reactivity of participating amino and carbonyl moieties [20]. The amine at C5 of the pyrimidine is generally more nucleophilic and will attack the more electrophilic carbonyl moiety. This is an important consideration when determining whether the desired functionalization of the pteridine is at C6 or C7 position under neutral conditions. The nucleophilicity of the amino group can be influenced by pH (Figure 3), and under strongly acidic conditions the C6 amine will initialize condensation by attacking the more reactive carbonyl to synthesize 7- substituted pterin as explored by Purrmann [21,22]. Regiospecific synthesis of this isomer has also been observed in condensation reactions that incorporate hydrazine to drive selectivity [23].

Specific regioisomers can also be isolated without modification of C5 pyrimidine nucleophilicity. The addition of NaHSO_3_ forms sulfite adducts that differ between 6- and 7- substituted isomers. These adducts can then be separated by their differences in solubility. Regioisomers of alkyl substituted pterins were separated by this method to isolate 6- substituted pterin with almost 99% isomer purity [24]. Additionally, NaHSO_3_ in combination with NaHCO_3_ has been shown to drive regioselectivity in Gabriel–Isay condensations that include hemiacetal functionality in the dicarbonyl (Figure 2) yielding tricyclic pyranopterin derivatives analogous to what is coordinated in the molybdenum cofactor. The use of 2,5,6-triamine-3,4-dihydropyrimidin-4-one sulfate without the addition of NaHSO_3_ or NaHCO_3_ protonates the amine at the C5 position, which yields the 7- substituted isomer as the major product. Adding NaHSO_3_ helps to neutralize the sulfate and free the C5 amine on the pyrimidine for nucleophilic attack and produce the 6- substituted isomer. Similar reactions performed with an analogous diketo ester were unable to separate 6- and 7- pterin isomers via sulfite adduct precipitation [20], which suggests this method of isolation to be system specific. Pyranopterin synthesis with Na_2_SO_3_·7H_2_O has also been utilized in a synthetic procedure to realize a dithiolene protected molybdopterin analogue of dephospho Form A, an oxidative degradation product of molybdenum cofactor [25].

### 2.2. Viscontini Reaction

The Viscontini reaction is like the Gabriel–Isay condensation, however, 1,2-dicarbonyl species are substituted for sugar derived α-oxo oximes to produce a regioselective product (Figure 3). Viscontini reported the synthesis of D-neopterin and L-monapterin by condensation reactions between 2,5,6-triamine-3,4-dihydropyrimidin-4-one and pentose phenylhydrazone derivatives under slightly acidic conditions [26]. Regioselectivity is achieved via Amadori rearrangement, which provides a ketone for the initial nucleophilic attack of the more active C5 pyrimidine amine. This method can also be utilized to incorporate pyran ring functionality to afford tricyclic pyranopterin. Clinch and coworkers reported the synthesis of cyclic pyranopterin monophosphate, a biosynthetic intermediate in the molybdenum cofactor pathway, which includes a Viscontini reaction with D-galactose phenylhydrazone [27]. A crystal structure for a Boc-protected derivative of this pyranopterin has also been reported [28].

### 2.3. Timmis Reaction

The disadvantage of isomer formation in pteridine synthesis can be avoided by condensing a 5-nitroso-6-aminopyrimidine with a methylene unit, more commonly known as the Timmis reaction (Figure 4) [29]. The methylene group can be activated with carbonyl functionality, and like the Gabriel–Isay condensation and Viscontini reaction, the Timmis reaction also initializes with a condensation between amino and carbonyl moieties usually under basic conditions. The preceding steps of the mechanism differ in that the 5-nitroso group undergoes a condensation with the methylene carbon to afford a regioselective product. The Timmis reaction has also been modified to incorporate Diels–Alder type condensation to complete pyrazine formation with [4 + 2] cycloaddition reactions. Synthesizing an oxadiazinone from the Timmis reaction, a dienophile can be incorporated to complete pyrazine cyclization via an intermolecular reaction [30]. An intramolecular Diels–Alder example from a Timmis reaction product has also been reported to synthesize a pterin derivative [31].

### 2.4. Polonovski–Boon Cyclization

Pterin has three possible redox states: fully oxidized, semi-reduced dihydropterin and fully reduced tetrahydropterin (Figure 4). These oxidation states are interconverted by 2 e^−^ and 2 H^+^ redox reactions that are integral to the function of redox active of biologically relevant pterins like biopterin and neopterin. The Polonovski–Boon cyclization provides a regiospecific method for synthesizing semi-reduced dihydropterin derivatives (Figure 5) [32,33]. The chlorinated carbon is activated by the strongly electron withdrawing nitroso group to facilitate nucleophilic attack of the amine. This will become the N8 of the semi-reduced pteridine once cyclization is completed. The nitroso group is then reduced to an amine and a condensation reaction with the ketone finishes the hetero-cyclic addition in a similar manner to previously discussed condensation methods. Reduced pterins are readily oxidized by strong oxidizing agents to yield fully oxidized pterin if desired [34,35].

### 2.5. Taylor Method

Unlike the previously described reactions, the Taylor method constructs a pyrimidine ring on a pyrazine frame by taking advantage of amine sources such as guanidine, urea or thiourea (Figure 6) [36,37]. The pyrimidine is formed by an insertion reaction between an amine and the -CN fragment of the pyrazine ring. The Taylor method can be limited by the functionality that can be imposed on the pyrazine ring, however, the source of the nitrogen for pyrimidine formation can be derived from the pyrazine starting material, amine or both. This method has been used in the synthesis of many biologically relevant pterin and pteridine compounds [38,39,40,41].

### 2.6. Sonogashira Coupling

The Sonogashira coupling is one of the most common methods for alkynylation of aryl or alkenyl halides [42]. This Pd(II) catalyzed cross coupling reaction forms a C-C bond between terminal alkyne and halide carbons (Figure 7). Any pterin used in a Sonogashira coupling reaction towards the synthesis of molybdenum cofactor needs to be selectively halogenated at the C6 position to ensure the product is of the correct regioisomer. For example, 6-chloropterin [43] has been modified to 2-pivaloyl-6-chloropterin [38] and used in such reactions with various terminal alkynes [44]. There is much interest in the alkynylation of pterin as the alkyne group can be readily reacted with polysulfide Mo compounds to produce small molecule mimics of molybdenum cofactor [44,45,46,47,48]. Additionally, the alkyne can be reduced to a protected dithiolene unit with the addition of a dithiocarbonate [49]. Pyranopterin functionality can be designed into these mimics when careful consideration is given to the starting alkyne. The addition of a primary alcohol α to the terminal alkyne can lead to pyranopterin formation after a dithiolene unit is appended to the cross-coupling product. The added “kink” from reducing the alkyne to an alkene places the primary alcohol closer to the C7 of the pteridine ring for possible pyran cyclization to occur. This has been observed in the crystal structure of a molybdenum cofactor mimic containing a completed pyranopterin system [47].

### 2.7. Aminopyrazine Pathway

Klewe et al. reported an adjustment to the Taylor method with an aminopyrazine-based pathway for the total synthesis for racemic dephospho Form A [50]. Initially, the target was the synthesis of 6-iodopterin (Figure 8) with the Taylor method and guanidine as the nitrogen source to complete the pyrimidine ring. The larger halogen present in 6-iodipterin would make a Sonogashira coupling more efficient, however, less than desirable yields of the iodine substituted pterin prompted further investigation into alternative approaches. Condensation reactions between ethoxycarbonylisothiocyate (SCNCO2Et) and amino-substituted heterocycles have been used to synthesize precursors to pteridine derivatives [51]. The resulting thiourea functionality can be activated with 1-3thyl-3-(3-dimethylaminopropyl)carbodiimide (EDCI) and hexamethyldisilane (HMDS) as the amine source to yield the desired pyrimidine ring [52]. After a Sonogashira coupling between a TBDPS-protected alkyne and methyl-3-amino-6-iodiopyrazein-2-carboxylate, the 3-amino group can be converted to thiourea and activated to complete pteridine cyclization (Figure 9). In the present case, the protecting groups of rac-dephospho Form A were removed under basic conditions, the deprotected product precipitated with EtOH and the desired product protonated and isolated with cation exchange resin in good yield.

## 3. Discerning between C6 and C7 Substituted Pterins

The possibility of synthesizing a mixture of regioisomers when working with asymmetric building blocks for pterin means that differentiating between C6 and C7 substituted isomers is crucial. Although it has been detailed how to selectively synthesize a desired pterin isomer above, some of these methods are shown to be system specific and should be taken as a general guide. Each system is different and therefore it is important to be able to readily identify a mixture of isomers and their relative abundance as conditions are optimized for a reaction.

The difference between pterin regioisomers can be easily observed from the ^1^H NMR spectrum of even a crude reaction mixture. The relative amount of C6 and C7 substituted isomers can be readily determined in a mixture of isomers based on the integral of the resonance for the proton on the unsubstituted carbon. This has been utilized to determine the relative yield of major and minor products for the condensation reaction shown in Figure 2. The asymmetric nature of the pyrimidine starting material provides slightly different chemical environments between C6 and C7 protons, which results in different shifts between the two isomers. The major product substituted at C6 shows a C7 proton resonance at 8.85 ppm, while the minor product substituted at C7 shows a C6 proton resonance at 8.56 ppm. These two peaks can be integrated to quickly determine the relative amounts of each in the reaction mixture. In general, it is expected that the C6 substituted product will have a C7 proton resonance that is slightly further downfield due to its relative proximity to the electron withdrawing nitrogen of the pyrimidine ring. 

Additionally, similar compounds can be distinguished by their absorption and emission spectra. This method is less precise than using NMR spectra, but it can still be useful when different isomers have significantly different electronic spectra. A set of model pyranopterin compounds synthesized using Gabriel–Isay condensation with 2,3-dimainopyridine (Figure 10) exhibits different absorption and emission spectra [20]. Both isomers were synthesized in equal amounts and chromatographically separated. Isomeric pyridine derivatives were different colors with different absorbance and emission peaks. Compound **1** was yellow with absorbance peaks at 371 and 389 nm and an emission peak at 423 nm, while **2** is yellow orange with absorbance peaks at 368 and 385 and an emission peak at 420 nm. The application of using electronic spectra to differentiate between pterin isomers may be somewhat limited, but utility may still be found in niche systems.

## 4. Impact of Pyran Cyclization on Redox Chemistry

One of the fundamental roles of pterin in biology is to participate in redox chemistry. The multiple redox states of pterin, as described in Figure 4, allow for 4 e^−^, 4 H^+^ redox transfer to occur over sequential redox steps. The redox chemistry of pterin and their relevance to enzyme activity are well documented, [53] however, there is not much discussion on how the cyclization of the pyran ring of pyranopterin influences the redox potential of the system. The open or closed nature of the pyran ring of pyranopterin has structural implications which can impose electronic perturbations on the organic framework and, in cases of metal coordination, the metal center. 

A simplified model pyranopterin system containing quinoxaline, dithiolene and pyran moieties has been synthesized using a Gabriel–Isay condensation between keto-esters and o-phenylaminediamine (Figure 11) [20]. The open ring compound can be readily closed by treatment with the benzyl chloroformate in very good yields. Additionally, the closed ring was synthesized directly with microwave irradiation in the presence of montmorillonite in modest yields. The open and closed ring nature of the pyran moiety of these compounds has been confirmed, respectively, with crystal structures. Pyran ring closure produces a more planar compound as the extra ring in the system eliminates rotation between quinoxaline and dithiocarbamate moieties, whereas the ring open form can freely rotate. The electrochemical properties of these compounds were explored with cyclic voltammetry and exhibit a significant difference in redox potential based on the state of the pyran ring. Both compounds show irreversible couples at room temperature, however, the reduction potential for the open form (−1530 mV) is roughly 300 mV more positive than the closed form (−1835 mV) [20]. Closing the pyran ring creates a completed conjugated system across the entire molecule, whereas opening the pyran ring interrupts electronic communication between the quinoxaline and dithiolene subunits. The difference in conjugation between these compounds could be driving the difference in redox potential, as a more delocalized system increases the difficulty of pyranopterin reduction.

The importance of the pyran ring of pyranopterin ligands is relevant to understanding the role of the MPT ligand in biological systems. Model compounds containing Mo coordinated by tris(3,5-dimethyl-1-pyrazolyl)borate (Tp*) and model MPT ligands (Figure 5) have been reported and demonstrate the importance of pyran cyclization to intraligand electronic communication [54,55]. In both instances pyran cyclization is modulated by the presence of a hydroxyl group on the carbon α to the dithiolene unit of the model ligand. The presence of a hydroxyl group allows for pyran cyclization to occur and more intense intraligand charge transfer (ILCT) bands are observed between 450 to 500 nm when compared to their methylated ring open derivatives. The increased intensity of this ILCT band is attributed to more facile charge transfer between pyrazine and dithiolene moieties which is produced by better electronic coupling between these units by pyran cyclization. The impact of ring open and ring closed forms on the reduction potential of the Mo center has also been probed with cyclic voltammetry. For both sets of model compounds, the redox potential for the Mo (V/IV) couple is more positive for compounds with pyran ring functionality. The redox couple observed for **3** is 115 mV more positive than **4**, whereas **5** is only 54 mV more positive than **6** [54,55]. The differences in reduction potentials observed for both ligand and metal-based redox processes indicates that pyran cyclization of the pyranopterin ligands can serve to electronically tune and modulate reactivity. It is also of interest to note that the pyran cyclization facilitates metal reduction, but increases the difficulty for ligand-based reduction as discussed above.

## 5. Pterin as Fluorophores

The molybdopterin (MPT) ligand of molybdenum cofactor is known to be fluorescent due to its core conjugated pyranopterin system. Pterin molecules typically exhibit fluorescence and substituting with various functional groups on the pterin can alter the emission energy. This useful property of pterins can see them utilized as fluorophores [56,57] and easily detected as biomarkers for viral infection, inflammation and cancer [58]. It was also reported that the fluorescence of red blood cells in tunicate, *Cnemidocarpa irene,* can be attributed to a pterin derivative [59]. The compounds highlighted in Figure 2 and Figure 10 also fluoresce similarly to that of oxidized MPT. Photophysical studies on the properties of indolylpterin (Figure 6) highlight the effects that solvent polarity has on the fluorescent properties pterin derivatives [60]. In nonpolar aprotic solvents (toluene, n-hexanes and CCl_4_) the intensity of the emission band is greater when compared to that in more polar aprotic solvents (ethyl acetate, acetonitrile and THF). In a polar protic solvent, like methanol, the emission band is greatly reduced, and fluorescence is almost lost. The fluorescence of indolylpterin is attributed to an excited state intramolecular proton transfer (ESIPT) between pyrazine and pyrrole rings, which is inhibited by the hydrogen bonding interactions with the solvent. Solvents with stronger hydrogen bond accepting properties will make ESIPT more difficult and result in a lower intensity of emission. In the case of methanol, the hydrogen bonding interaction between the solvent and the indolylpterin is so great that fluorescence is almost completely lost. The indole-free 7-methyl derivative of indolylpterin does not have its fluorescence inhibited by polar protic solvents because ESIPT does not occur.

## 6. Computational Studies

Theoretical methods are an important tool in understanding the properties and reactivity of small organic molecules. These uses range from determining the electronic structure of a single organic compound to modeling entire reaction mechanisms. The ability to model closed and open shell systems give *ab initio* methods the flexibility to probe a wide range of pterins in various oxidation states. This section focuses on theoretical work describing the stability of various pterin radicals, the impacts of oxidation on tautomerization and fluorescence.

Detailed density functional theory (DFT) studies on neutral pterin radicals have also been performed to further elucidate the electronic structure and identify the most stable tautomeric form. Gas and solvation models based on density (SMD) calculations were performed on five neutral homolytically cleaved pterin radicals (Figure 7) with open shell calculations to understand how the location of the radical affects spin density and delocalization [61]. Carbon-based radicals showed no electron delocalization, while nitrogen-based radicals exhibited significant electron delocalization. Optimized geometries of the neutral radicals in Figure 7 show that carbon-based radicals exhibit non-planar geometries around N2′ which is a result of weak electron delocalization. All nitrogen-based radicals show exclusively planar structures. Unsurprisingly, carbon-based radicals are much less favorable than neutral radicals centered on nitrogen atoms. In the gas phase the order of radicals from most to least stable is: N2′a < N3 < N2′b < C7 < C6, and in the condensed phase the order remains mostly unchanged, however, N3 is more favorable than N2′a. This change is attributed to increased polarity of the carbonyl bond brought by implicit solvation which stabilizes the N3 radical.

Folic acid (FA) is the pterin containing derivative of vitamin B9 that is utilized in FA coenzyme to catalyze reactions that participate in the metabolism of nucleic acids and proteins [62]. Epidemiological studies on folate supplements suggest that risk for pancreatic and breast cancer may be mitigated by free-radical scavenging and antioxidant processes [63]. The pterin moiety of FA can theoretically tautomerize between lactam and lactim forms (Figure 8), however the lactam form is much more favored. It is suggested that the unfavored lactim isomer can undergo oxidation and possibly provide insight into the antioxidant properties of folate [64]. DFT calculations were performed on 6-methylpterin as an analogue for FA to find the connection between lactam/lactim tautomerization and oxidation [65]. The transition state (TS) between lactim and lactam isomers was calculated to show that two intervening water molecules are required to afford the lowest activation energy for the tautomerization. Oxidation reactions of 6-methylpterin were explored in the presence of the intervening water and OX^•^ (HO^•^, Cl_3_C-O_2_^•−^, N_3_^•^ and SO_4_^•−^) as oxidants. In the first step the fleeting lactim tautomer is formed via proton transfers between the water and pterin. Next, the pterin donates an electron to the oxidant and the radical nature is transferred to N4 of 6-methylpterin. This indicates that oxidation occurs via a single electron transfer between pterin and oxidant, as no radical character is observed in water molecules that participate in oxidation, ruling out a hydrogen atom transfer mechanism.

Oxidized pterins are known to be highly fluorescent [66] and act as photosensitizers to form triplet excited states [67] which results in the production of singlet oxygen generation in high quantum yields [68]. Triplet state pterins can react with other pterins to form reactive cationic and anionic radicals in a reaction related to the mechanism of a type II photosensitizer. Anionic radical pterins can also be formed by the presence of triplet pterin with an electron donor and react with molecular oxygen to form a superoxide-anion radical (Equations (1)–(4)). Pterins in alkaline conditions are known to participate in direct electron transfer with O_2_, whereas the neutral form cannot undergo such a reaction. The effectiveness of a pterin to act as a photosensitizer is dependent on the substituent at the C6 position [69]. The photophysical and photochemical characteristics for several pterin compounds (Figure 9) were calculated for both acidic and basic forms to establish how the functionality of C6 and C7 influences the photochemical properties. Geometries of these compounds were optimized to predict the electronic absorption with time dependent (TD)-DFT calculations, vertical ionization energy and electron affinity energy. The wavelength of phosphorescence was predicted by taking the adiabatic difference in energy of the optimized ground state and T1 state, and agree well with experimental data [70]. Pterins with electron withdrawing substituents have larger ionization potentials which correlates well with their decreased ^1^O_2_ quenching rate constants. It was also determined that the quantum yield of ^1^O_2_ generation depends on the rate of energy transfer to O_2_ and the quantum yield of the formation of the triplet state [70].
^3^Ptr* + Ptr → Ptr^•−^ + Ptr^•+^(1)
^3^Ptr* + O_2_ → ^1^O_2_ + Ptr(2)
^3^Ptr* + D → Ptr^•−^ + D^•+^(3)
Ptr^•−^ + O_2_ → Ptr + O_2_^•−^(4)

As discussed previously, the fluorescent properties of pterin can be quenched via excited-state proton transfers (ESPT), and computational methods have been utilized to explore this mechanism for quenching. Pterins are known to exhibit pH-dependent fluorescence quenching in the presence of various hydrogen acceptors such as phosphate and acetate [71]. Pterins in acidic solutions are quenched in the presence of hydrogen acceptors, while the fluorescence of a pterin under basic conditions is unchanged. The influence of ESPT on pterin fluorescence was studied using condensed phase DFT and TD-DFT approaches in the presence of an acetate anion [72]. The electrostatic potential surfaces for the acidic and basic forms of pterin show that the hydrogen of the amino and imino groups for the acidic form of a pterin are more positive and provide better hydrogen bonding for acetate (Figure 10). The lack of sufficient hydrogen bonding between the acetate and basic form of pterin makes proton transfer in the ground and excited state highly unlikely. The strong interaction between acetate and the acidic form of pterin produces a favorable ESPT in which a proton of N5 is donated to acetate anion. The potential energy curves and surfaces modeling this transfer indicate that the proton transfer is specific to the N5 site, as primary amines are more acidic than secondary amines. Excited state calculations for the HOMO → LUMO transition show electron density shifting from N5 to N1 upon excitation. This loss of electron density at N5 makes that amine even more acidic which enhances the hydrogen bonding interactions and facilitates ESPT leading to fluorescence quenching.

## 7. Summary

This overview of pterin chemistry illustrates a variety of the strategies that have been used to synthesize pterins and how these methods can be utilized in the synthesis of biologically relevant compounds. Pterin synthesis can be realized by different condensation reactions utilizing either pyran or pyrimidine building blocks. Coupling an appropriately substituted alkyne on the C6 position of pterin opens the possibility to forming dithiolene and pyran moieties present in the molybdenum cofactor. Additionally, the fluorescent and antioxidant properties of pterins have been presented and augmented by conclusions obtained by theoretical calculations. Pterins are strongly fluorescent molecules that can be quenched via ESPT which can occur by both inter- and intramolecular processes, and pterin can act as an electron donor to participate in oxidation reactions, serving as the reductant. This review highlights a few examples of the diverse chemistry offered by pterins and their importance in biological systems.

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
