# Peer review of "Synthesis, Redox and Spectroscopic Properties of Pterin of Molybdenum Cofactors"

_molecules, 2022, doi:10.3390/molecules27103324_

Round 1

Reviewer 1 Report

The manuscript for a review article by Colston and Basu was submitted to be part of the special issue "State-of-the-Art in Molybdenum Cofactor Research" for which is beyond any doubt very suitable since it covers exciting chemistry which is at the very heart of these cofactors. The authors detail every aspect of pterin chemistry which is a component of the molypdopterin ligand, an integral part of the cofactors. They have included synthesis as well as characterisation and what the latter implies for the operation of the natural template. The review appears to be comprehensive. It has a very good balance between breadth and depth. It is, therefore, recommended to publish it in Molecules as part of the special issue it was submitted for.

A few minor issues should be taken care of, though:

In the abstract the term pterin should be better used in plural form (as in the very first word), or at least the text should be checked for subject verb agreement. The latter more generally should be carefully checked throughout.

The text must also be checked for some few typos, e.g. natute, moeity, pyrazein, pteirn, inodole,

It would be very valuable if the group tolerance (R, R’) could be indicated in schemes 4 to 10.

For the ligand it is stated that the open form has a more positive potential than the pyran form (lines 254-256). For the complex it is stated that the pyran form has a more positive potential (lines 275-277). There appears to be a contradiction. It may be that the second statement is not precise enough regarding to which species this applies. It might have to be rephrased to become unambiguous.

Author Response

Response to Reviewers of Synthesis, Redox, and Spectroscopic Properties of Pterin of Molybdenum Cofactor

Reviewer 1:
The grammatical changes suggested have all be implemented in the current form of the manuscript. Pterin is properly pluralized throughout the manuscript and all typographical errors have been corrected.

The tolerance groups of Schemes 3 – 6 have been updated to give the reader a better idea of the types of substituents that have been utilized for these reactions. Schemes 7 has been updated to highlight specific cross coupling reactions that have been performed to create functionalized pterins at the 6 position. Additionally, Scheme 8 has been removed as it was determined to be redundant upon reflection. Figure 4 and Scheme 11 succinctly illustrate pyranopterin ring closure with organic and inorganic example compounds.

The confusion regarding redox potentials of open and closed form pterin has been addressed. Those statements seemed contradictory because they are discussing different types of redox processes. The first instance in which pyran formation makes reduction more negative is in the context of a ligand-based reduction. The next statement about ring closure making reduction easier describes a Mo-based reduction potential. A statement has been added to clarify the different effects that pyran ring closure has on ligand and metal-based redox properties.

We believe that the changes prescribed by the reviewers has made the newest edition of the manuscript a stronger and more clear piece for the readership of Molecules.

Author Response

Response to Reviewers of Synthesis, Redox, and Spectroscopic Properties of Pterin of Molybdenum Cofactor

Reviewer 2:

All the grammatical changes suggested by the reviewer have been applied to the newest edition of this manuscript. Additionally, references have been added to the sections where changing redox potentials are discussed.

Instances in which computational theory is described in the text have also been removed.

The first statement in the introduction has been clarified to include “All phyla of life” instead of “All life”.

An additional figure has been added to illustrate Molybdenum cofactor to give the reader more context as to how pterin relates to the biological system, as suggested by the reviewer.

Figure 6 (Now 7 in the current version) has been corrected as N3 was incorrectly labeled as “N5”.

We believe that the changes prescribed by the reviewers has made the newest edition of the manuscript a stronger and more clear piece for the readership of Molecules.